# Temporal Difference Weighted Ensemble for Reinforcement Learning

## Abstract

Combining multiple function approximators in machine learning models typically leads to better performance and robustness compared with a single function. In reinforcement learning, ensemble algorithms such as an averaging method and a majority voting method are not always optimal, because each function can learn fundamentally different optimal trajectories from exploration. In this paper, we propose a Temporal Difference Weighted (TDW) algorithm, an ensemble method that adjusts weights of each contribution based on accumulated temporal difference errors. The advantage of this algorithm is that it improves ensemble performance by reducing weights of Q-functions unfamiliar with current trajectories. We provide experimental results for Gridworld tasks and Atari tasks that show significant performance improvements compared with baseline algorithms.

## 1 Introduction

Using ensemble methods that combine multiple function approximators can often achieve better performance than a single function by reducing the variance of estimation (Dietterich (2000); Kuncheva (2014)). Ensemble methods are effective in supervised learning, and also reinforcement learning (Wiering & Van Hasselt (2008)). There are two situations where multiple function approximators are combined: combining and learning multiple functions during training (Freund & Schapire (1997)) and combining individually trained functions to jointly decide actions during testing (Breiman (1996)). In this paper, we focus on the second setting of reinforcement learning wherein each function is trained individually and then combined them to achieve better test performance.

Though there is a body of research on ensemble algorithms in reinforcement learning, it is not as sizeable as the research devoted to ensemble methods for supervised learning. Wiering & Van Hasselt (2008) investigated many ensemble approaches combining several agents with different value-based algorithms in Gridworld settings. Faußer & Schwenker (2011; 2015a) have shown that combining value functions approximated by neural networks improves performance greater than using a single agent. Although previous work dealt with each agent equally contributing to the final output, weighting each contribution based on its accuracy is also a known and accepted approach in supervised learning (Dietterich (2000)).

However, unlike supervised learning, reinforcement learning agents learn from trajectories resulting from exploration, such that each agent learns from slightly different data. This characteristic is significant in tasks with high-dimensional state-space, where there are several possible optimal trajectories to maximize cumulative rewards. In such a situation, the final joint policy function resulting from simple averaging or majority voting is not always optimal if each agent learned different optimal trajectories. Furthermore, it is difficult to decide constant weights of each contribution as it is possible that agents with poor episode rewards have better performance in specific areas.

In this paper, we propose the temporal difference weighted (TDW) algorithm, an ensemble method for reinforcement learning at test time. The most important point of this algorithm is that confident agents are prioritized to participate in action selection while contributions of agents unfamiliar with the current trajectory are reduced. To do so in the TDW algorithm, the weights of the contributions at each Q-function are calculated as softmax probabilities based on accumulated TD errors. Extending an averaging method and a majority voting method, actions are determined by weighted average or voting methods according to the weights. The advantage of the TDW algorithm is that arbitrary

training algorithms can use this algorithm without any modifications, because the TDW algorithm only cares about the joint decision problem, which could be easily adopted in competitions and development works using reinforcement learning. In our experiment, we demonstrate that the TDW retains performance in tabular representation Gridworld tasks with multiple possible trajectories, where simple ensemble methods are significantly degraded. Second, to demonstrate the effectiveness of our TDW algorithm in high-dimensional state-space, we also show that our TDW algorithm can achieve better performance than baseline algorithms in Atari tasks (Bellemare et al. (2013)).

## 2 RELATED WORK

Ensemble methods that combine multiple function approximators during training rather than evaluation have been studied in deep reinforcement learning. Bootstrapped deep Q-network (DQN) (Osband et al. (2016)) leverages multiple heads that are randomly initialized to improve exploration, because each head leads to slightly different states. Averaged-DQN (Anschel et al. (2017)) reduces the variance of a target approximation by calculating an average value of last several learned Q-networks. Using multiple value functions to reduce variance of target estimation is also utilized in the policy gradients methods (Fujimoto et al. (2018); Haarnoja et al. (2018)).

In contrast, there has been research focused on joint decision making in reinforcement learning. Using multiple agents to jointly select an action achieves better performance than a single agent (Wiering & Van Hasselt (2008); Faußer & Schwenker (2015a; 2011)). However, such joint decision making has been limited to relatively small tasks such as Gridworld and Maze. Therefore, it is not known whether joint decision making with deep neural networks can improve performance in high-dimensional state-space tasks such as Atari 2600 (Bellemare et al. (2013)).

Doya et al. (2002) proposes a multiple mode-based reinforcement learning (MMRL), a weighted ensemble method for model-based reinforcement learning, which determines each weight based by using prediction models. The MMRL gives larger weights to reinforcement learning controllers with small errors of special responsibility predictors. Unlike MMRL, our method does not require additional components to calculate weights.

Our method is not the first one to use TD errors in combining multiple agents. Ring & Schaul (2011) proposes a module selection mechanism that chooses the module with smallest TD errors to learn current states, which will eventually assign each module to a small area of a large task. As a joint decision making method, a selective ensemble method is proposed to eliminate agents with less confidence at the current state by measuring TD errors (Faußer & Schwenker (2015b)), which is the closest approach to our method. This selection drops all outputs whose TD errors exceeds a threshold, which can be viewed as a hard version of our method that uses a softmax of all weighted outputs instead of elimination. The threshold is not intuitively determined. Because the range of TD errors varies by tasks and reward settings, setting the threshold requires sensitive tuning.

## 3 BACKGROUND

### 3.1 REINFORCEMENT LEARNING

We formulate standard reinforcement learning setting as follows. At time $t$, an agent receives a state $s_t \in \mathbb{S}$, and takes an action $a_t \in \mathbb{A}$ based on a policy function $a_t = \pi(s_t)$. The next state $s_{t+1}$ is given to the agent along with a reward $r_{t+1}$. The return is defined as a discounted cumulative reward $R_t = \sum_{i=t}^{T} \gamma^{i-t} r(s_i, a_i)$, where $\gamma \in [1, 0]$ is a discount factor. The true value of taking an action $a_t$ at a state $s_t$ is described as follows:

$$Q_\pi(s_t, a_t) = \mathbb{E}[R_t | s_t, a_t]$$

where $Q_\pi(s_t, a_t)$ is an action-value under the policy $\pi$. The optimal value is $Q_*(s_t, a_t) = \max_\pi Q_\pi(s_t, a_t)$. With such an optimal Q-function, optimal actions can be determined based on the highest action-values at each state.

DQN (Mnih et al. (2015)) is a deep reinforcement learning method that approximates an optimal Q-function with deep neural networks. The Q-function $Q(s_t, a_t | \theta)$ with a parameter $\theta$ is approximated by a Q-learning style update (Watkins & Dayan (1992)). The parameter $\theta$ is learned to minimize

squared temporal difference errors.

$$L(\theta) = \mathbb{E}_{s_t, a_t, r_{t+1}, s_{t+1}}[(Q(s_t, a_t|\theta) - y_t)^2] \tag{1}$$

where $y_t = r_{t+1} + \gamma \max_a Q(s_{t+1}, a|\theta')$ with a target network parameter $\theta'$. The target network parameter $\theta'$ is synchronized to the parameter $\theta$ in a certain interval. DQN also introduces use of the experience replay (Lin (1992)), which randomly samples past state transitions from the replay buffer to compute the squared TD error (1).

## 3.2 ENSEMBLE METHODS

Assume there are $N$ sets of trained Q-function $Q(s, a|\theta_i)$ where $i$ denotes an index of the function. The final policy $\pi(s_t)$ is determined by combining the $N$ Q-functions. We formulate two baseline methods commonly used in ensemble algorithms: Average policy and Majority Voting (MV) policy (Faußer & Schwenker (2011; 2015a); Kuncheva (2014)).

Majority Voting (MV) policy is an approach to decide the action based on greedy selection according to the formula:

$$\pi(s_t) = \arg\max_a \sum_i^N v_i(s_t, a) \tag{2}$$

where $v_i(s, a)$ is a binary function that outputs 1 for the most valued action and 0 for others:

$$v_i(s, a) = \begin{cases} 1 & (a = \arg\max_{a'} Q(s, a'|\theta_i)) \\ 0 & (otherwise) \end{cases} \tag{3}$$

Contributions of each function to the final output are completely equal.

Average policy is a method that averages all the outputs of the Q-functions, and the action is greedily determined:

$$\pi(s_t) = \arg\max_a \frac{1}{N} \sum_i^N Q(s_t, a|\theta_i) \tag{4}$$

Averaging outputs from multiple approximated functions reduces variance of prediction. Unlike MV policy, Average policy leverages all estimated values as well as the highest values.

## 4 TEMPORAL DIFFERENCE WEIGHTED ENSEMBLE

In this section, we explain the TDW ensemble algorithm that adjusts weights of contributions based on accumulated TD errors. The TDW algorithm is especially powerful in the complex situation such as high-dimensional state-space where it is difficult to cover whole state-space with a single agent. Section 4.1 describes the error accumulation mechanism. Section 4.2 introduces joint action selection using the weights computed with the accumulated errors.

### 4.1 ACCUMULATING TD ERRORS

We consider that a squared TD error $\delta_t^i = (Q(s_t, a_t|\theta_i) - r_{t+1} - \gamma \max_a Q(s_{t+1}, a|\theta_i))^2$ fundamentally consists of two kinds of errors:

$$\delta_t^i = \delta_t^{i,p} + \delta_t^{i,u} \tag{5}$$

where $\delta^p$ is a prediction error of approximated function, and $\delta^u$ is an error at states where the agent rarely experienced. In a tabular-based value function, $\delta^p$ will be near 0 at frequently visited states. In contrast, $\delta^u$ will be extremely large at less visited states with both a tabular-based value function and a function approximator because TD errors are not sufficiently propagated such a state. There are two causes of unfamiliar states: (1) states are difficult to visit due to hard exploration, and (2) states are not optimal to the agent according to learned state transitions. For combining multiple agents at a joint decision, the second case is noteworthy because each agent may be optimized at different optimal trajectories. Thus, some of the agents will produce larger $\delta^u$ when they face such states as a result of an ensemble, and contributions of less confident agents can be reduced based on the TD error $\delta^u$.

---

**Algorithm 1** Temporal Difference Weighted Ensemble

---

**Require:** $\alpha$: a constant decay factor
**Require:** $\theta_1...\theta_N$: trained parameters
  Initialize $u_0^1 = 0, ..., u_0^N = 0$.
  **for** $t = 1, 2, \ldots, T$ **do**
    Receive state $s_t$ and reward $r_t$.
    **for** $i = 1, 2, \ldots, N$ **do**
      Calculate $u_t^i$ via (6).
    **end for**
    **for** $i = 1, 2, \ldots, N$ **do**
      Calculate $w_t^i$ via (7).
    **end for**
    Select action $a_t$ via (8) or (9).
  **end for**

---

To measure uncertainty of less confident agents, we define $u_t^i$ as a uncertainty of an agent:

$$u_t^i = \delta_{t-1}^i + \alpha u_{t-1}^i \tag{6}$$

where $\alpha \in [0, 1]$ is a constant factor decaying the uncertainty at a previous step. With a large $\alpha$, the uncertainty $u^i$ is extremely large during unfamiliar trajectories, which makes it possible to easily distinguish confident agents from the others. However, a trade-off arises when prediction error $\delta_p$ is accumulated for a long horizon, which increases correlation between agents.

## 4.2 ACTION SELECTION

To reduce contributions of less confident agents, each contribution at joint decision is weighted based on uncertainty $u_t^i$. Using the uncertainty $u_t^i$, a weight $w_t^i$ of each agent is calculated as a probability by the softmax function:

$$w_t^i = \frac{e^{-u_t^i}}{\sum_j^N e^{-u_t^j}} \tag{7}$$

When the agent has a small uncertainty value $u_t^i$, the weight $w_t^i$ becomes large.

We consider two weighted ensemble methods corresponding to the Average policy and the MV policy based on the weights $w_t^i$. As a counterpart of the Average policy, our TDW Average policy is as follows:

$$\pi(s_t) = \arg\max_a \sum_i^N u_t^i Q(s_t, a|\theta_i) \tag{8}$$

For the MV policy, TDW Voting policy is as follows:

$$\pi(s_t) = \arg\max_a \sum_i^N u_t^i v_i(s_t, a) \tag{9}$$

Unlike the averaging method, because TDW Voting policy directly uses probabilities calculated by (7), the correlation between agents can be increased significantly with large decay factor $\alpha$, leading to worse performance. Although these weighted ensemble algorithms are simple enough to extend to arbitrary ensemble methods, we leave more advanced applications for future work so that we may demonstrate the effectiveness of our approach in a simpler setting. The complete TDW ensemble algorithm is described in Algorithm 1.

## 5 EXPERIMENTS

In this section, we describe the experiments performed on the Gridworld tasks and Atari tasks (Bellemare et al. (2013)) in Section 5.2. To build the trained Q-functions, we used the table-based Q-learning algorithm (Watkins & Dayan (1992)) and DQN (Mnih et al. (2015)) with a standard model, respectively. In each experiment, we evaluated our algorithm to address performance improvements from there baselines as well as the effects of selecting the decay factor $\alpha$.

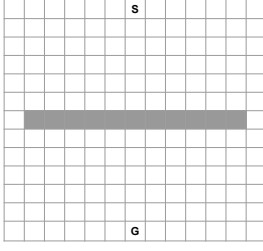
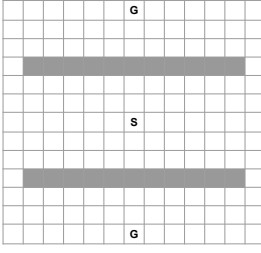

(a) Two-slit Gridworld          (b) Four-slit Gridworld

Figure 1: Gridworld environments. S denotes initial states. G denotes goals with $+100$ reward. Gray areas represents walls.

Table 1: Results for Gridworld tasks. The algorithms of best mean episode rewards are highlighted in bold font. The numbers in brackets show ranks based on significant difference computed with Welch's t-test ($p < 0.05$). The results of Single (mean) are the episode reward averaged over all single Q-functions.

| Method | Two-slit Gridworld | Four-slit Gridworld |
|---|---|---|
| Single (mean) | 97.43 | 98.07 |
| Average | 77.72 (6) | -9.57 (6) |
| TDW Average ($\alpha = 0.0$) | 89.04 (5) | 73.77 (5) |
| TDW Average ($\alpha = 0.2$) | 96.90 (2-4) | 85.73 (1-4) |
| TDW Average ($\alpha = 0.4$) | 96.94 (2-4) | 85.98 (1-4) |
| TDW Average ($\alpha = 0.6$) | 96.94 (2-4) | **86.22** (1-4) |
| TDW Average ($\alpha = 0.8$) | **97.00** (1) | 86.14 (1-4) |
| Majority Voting | 96.49 (6) | 36.51 (6) |
| TDW Voting ($\alpha = 0.0$) | 97.43 (1-5) | 94.32 (1-4) |
| TDW Voting ($\alpha = 0.2$) | **97.44** (1-5) | **94.57** (1-4) |
| TDW Voting ($\alpha = 0.4$) | 97.43 (1-5) | 94.45 (1-4) |
| TDW Voting ($\alpha = 0.6$) | 97.43 (1-5) | 94.42 (1-4) |
| TDW Voting ($\alpha = 0.8$) | 97.43 (1-5) | 94.15 (5) |

## 5.1 GRIDWORLD

### 5.1.1 EXPERIMENTAL SETUP

We first evaluated the TDW algorithms with a tabular representation scenario to show their effectiveness in the situation where it is difficult to cover a whole state-space with the single agent. We built two Gridworld environments as shown in Figure 1. Each environment is designed to induce bias of learned trajectories by setting multiple slits. As a result of exploration, once an agent gets through one of the slits to the goal, the agent is easily biased to aim for the same slit due to the max operator of Q-learning.

The state-representation is a discrete index of a table with size of $13 \times 13$. There are four actions corresponding to steps of *up*, *down*, *left* and *right*. If a wall exists where the agent tries to move, the next state remains the same as the current state. The agent always starts from S depicted in Figure 1. At every timestep, the agent receives a reward of $-0.1$ or $+100$ at goal states. The agent starts a new episode if either the agent arrives at the goal states or the timestep reaches 100 steps.

We trained $N = 10$ agents with different random seeds for $\epsilon$-greedy exploration with $\epsilon = 0.3$. Each training continues until 1M steps have been simulated. We set the learning rate to 0.01 and $\gamma = 0.95$. After training, we evaluated TDW ensemble algorithms for 20K episodes. As baselines, we also evaluate each single agent as well as ensemble methods of Average policy and MV policy for 20K episodes each.

Table 2: Evaluation results for the Atari tasks. The results of algorithms with the best mean episode rewards are highlighted in boldface. The numbers in brackets show ranks based on significant difference computed with Welch's t-test ($p < 0.05$). The results of Single (best) are the mean episode rewards of the best Q-functions.

| Method | Asterix | Beamrider | Breakout | Enduro | MsPacman | SpaceInvaders |
|---|---|---|---|---|---|---|
| Single (mean) | 3465.74 | 6616.89 | 307.99 | 582.38 | 1695.27 | 830.07 |
| Single (best) | 4022.90 | 8046.62 | 366.18 | 817.73 | 1776.74 | 1103.93 |
| Average | 6835.15 (1-3) | 11695.67 (5-6) | 404.21 (3-4) | 1400.98 (4) | 1734.89 (3-4) | 989.19 (5) |
| Weighted Average | 3950.10 (7) | 8126.58 (7) | 364.22 (6) | 793.06 (7) | **1852.79** (1) | 1092.32 (1-2) |
| TDW Average ($\alpha = 0.0$) | 6691.90 (4) | 10128.44 (5-6) | 418.05 (2) | 1441.50 (2) | 1815.54 (2) | 1076.21 (3-4) |
| TDW Average ($\alpha = 0.2$) | **6911.10** (1-2) | **12878.37** (1-2) | 403.43 (3-4) | **1445.44** (1) | 1711.20 (3-4) | 1030.01 (3-4) |
| TDW Average ($\alpha = 0.4$) | 6409.20 (5-6) | 11516.25 (3-5) | 362.21 (7) | 1330.12 (5-6) | 1698.05 (4-5) | 986.95 (6) |
| TDW Average ($\alpha = 0.6$) | 6907.55 (2-3) | 11386.13 (3-4) | 380.12 (5) | 1415.18 (3) | 1685.76 (6) | **1124.90** (1-2) |
| TDW Average ($\alpha = 0.8$) | 6472.25 (5-6) | 12661.50 (1-2) | **420.14** (1) | 1316.30 (5-6) | 1668.54 (7) | 902.91 (7) |
| Majority Voting | 4586.85 (6) | 9945.25 (6) | 344.30 (4) | 1204.10 (2) | 1542.78 (4) | 889.38 (7) |
| Weighted Voting | 4087.15 (7) | 8064.10 (7) | 364.91 (3) | 799.96 (7) | **1767.05** (1) | **1097.84** (1) |
| TDW Voting ($\alpha = 0.0$) | **6669.60** (1-2) | **11322.70** (1-2) | 340.21 (5) | 1149.08 (5) | 1651.31 (3) | 1038.13 (2) |
| TDW Voting ($\alpha = 0.2$) | 5565.75 (5) | 10982.07 (1-2) | 370.18 (2) | **1208.00** (1) | 1457.43 (5) | 928.29 (6) |
| TDW Voting ($\alpha = 0.4$) | 6210.80 (2-3) | 10790.61 (3) | 298.14 (7) | 1163.06 (3-4) | 1693.65 (2) | 1014.99 (3) |
| TDW Voting ($\alpha = 0.6$) | 5972.80 (4) | 10426.07 (4) | 336.81 (6) | 1172.65 (3-4) | 1392.62 (7) | 968.90 (4) |
| TDW Voting ($\alpha = 0.8$) | 6303.35 (1-3) | 10352.76 (5) | **410.29** (1) | 1141.04 (6) | 1410.55 (6) | 934.24 (5) |

### 5.1.2 RESULTS

The evaluation results on the Gridworld environments are shown in Table 1. Four-slit Gridworld is significantly more difficult than Two-slit Gridworld because each Q-function is not only horizontally biased, but also vertically biased. In both the Two-slit Gridworld and Four-slit Gridworld environments, the TDW ensemble methods achieve better performance than their corresponding Average policy and the MV policy baselines. Additionally, the results of both of the Average policy and the MV policy were worse than the single models. It should be noted that Average policy degrades original performance more than MV policy.

For the selection of the decay factor $\alpha$, a larger $\alpha$ tends to increase performance in TDW Average policy. In contrast, the larger $\alpha$ leads to poor performance in TDW Voting policy especially in Four-slit Gridworld. We believe that the large $\alpha$ significantly reduces contributions of most Q-functions, which would ignore votes of actions that would be the best in equal voting. In contrast, TDW Average policy leverages values of all actions, exploiting all contributions to select the best action.

### 5.2 ATARI

### 5.2.1 EXPERIMENTAL SETUP

To demonstrate effectiveness in high-dimensional state-space, we evaluated TDW algorithm in Atari tasks. We trained DQN agents across 6 Atari tasks (Asterix, Beamrider, Breakout, Enduro, MsPacman and SpaceInvaders) through OpenAI Gym (Brockman et al. (2016)). At each task, $N = 10$ agents were trained with different random seeds for neural network initialization, exploration and environments in order to vary the learned Q-function. The training continued until 10M steps (40M game frames) with frame skipping and termination on loss of life enabled. The $\epsilon$ of exploration is linearly decayed from 1.0 to 0.1 through 1M steps. The hyperparameters of neural networks are same as (Mnih et al. (2015)).

After training, evaluation was conducted with each Q-function, TDW Average policy, TDW Voting policy and the two baselines. We additionally evaluated weighted versions of the baselines whose Q-functions were weighted based on their evaluation performance. The evaluation continued for 1000 episodes with $\epsilon = 0.05$.

### 5.2.2 RESULTS

The experimental results are shown in Table 2. Interestingly, both of Average policy and MV policy improved performance from mean performance of single agents, though the simple ensemble algorithms had not been investigated well in the domain of deep reinforcement learning. In the games of Asterix, Beamrider, Breakout and Enduro, the TDW algorithms achieve additional performance

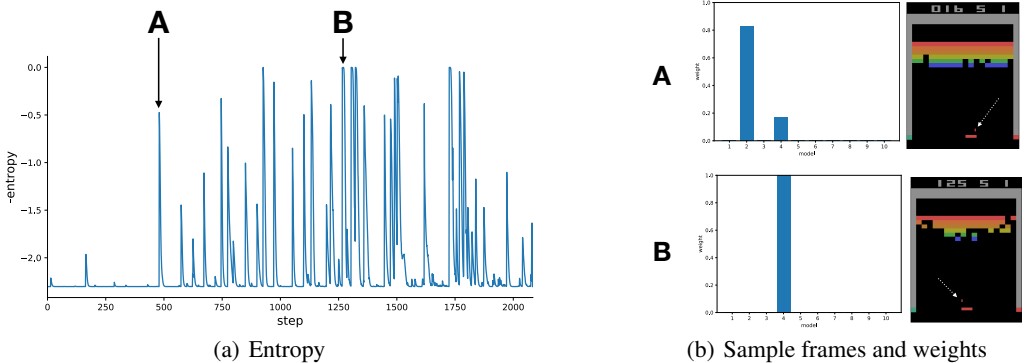

(a) Entropy  (b) Sample frames and weights

Figure 2: (a) Plots of entropies during a sample episode in Breakout. The $x$-axis represents episode steps. The $y$-axis represents negative entropies of the weights. (b) Bar graphs next to the game frames show the corresponding weights with the Q-function index on the $x$-axis. White arrows on the game frames show the ball positions and past trajectory.

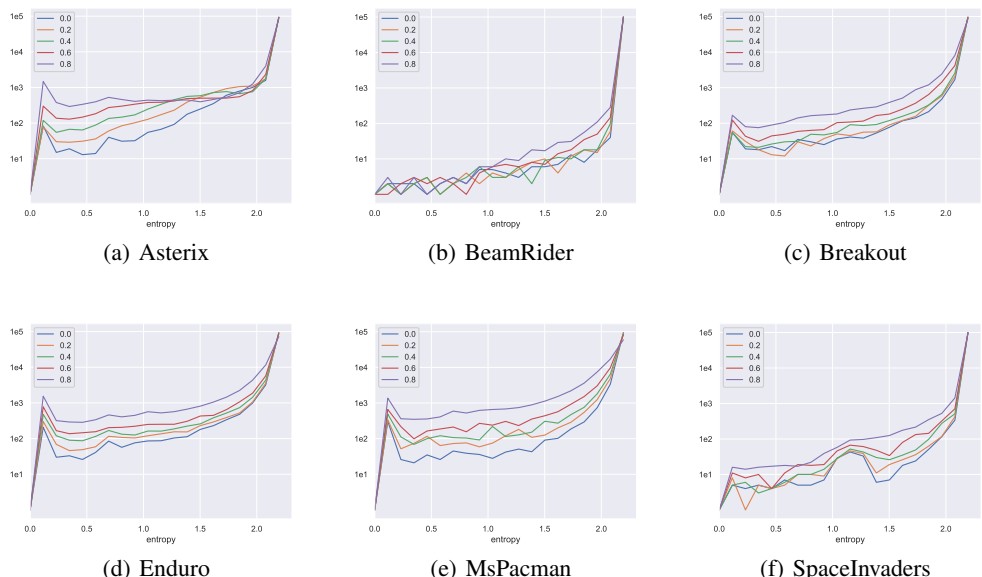

(a) Asterix  (b) BeamRider  (c) Breakout

(d) Enduro  (e) MsPacman  (f) SpaceInvaders

Figure 3: Plots of the number of observations with a certain entropy through 100K evaluation steps (TDW Average policy). $x$-axis represents entropy values with 20 bins. $y$-axis represents corresponding numbers in log scale.

improvements as compared with the non-weighted and weighted baselines. Even in MsPacman and SpaceInvaders, the TDW algorithms perform significantly better than non-weighted baselines and the best single models. In most of the cases, the globally weighted ensemble baselines performed worse than non-weighted versions. We believe this is because these globally weighted ensemble methods will ignore local performance, which is significant in high-dimensional state-space because it is difficult to cover all possible states with single models. The TDW algorithms with small $\alpha$ tend to achieve better performance than those with a large $\alpha$, which suggests that a significantly large $\alpha$ can increase correlation between Q-functions and reduce contributions of less confident Q-functions.

To analyze changes of weights through an episode, we plot entropies during a sample episode on Breakout (TDW Average policy, $\alpha = 0.8$) in Figure 2(a). If an entropy is low (high in negative scale), some of Q-functions have large weights, while others have extremely small weights. Extreme

low entropies are observed when the ball comes closely to the pad as shown in Figure 2(b) where the value should be close to 0 for the non-optimal actions because missing the ball immediately ends its life. It is easy for sufficiently learned Q-functions to estimate such a terminal state so that the entropy becomes low due to the gap between optimal Q-functions for the current states and the others. The entropies tend to be low during latter steps as there are many variations of the remaining blocks. We consider that the reason why the TDW algorithms with $\alpha = 0.8$ achieved the best performance in Breakout is that the large $\alpha$ value reduces influence of the Q-functions which cannot correctly predict values with unseen variations of remaining blocks. In contrast to Breakout where living long leads to higher scores, in SpaceInvaders we observe that the low entropies appear at dodging beams rather than shooting invaders, because shooting beams requires long-term value prediction which does not induce large TD errors. Therefore, performance improvements on SpaceInvaders are not significantly better than weighted baselines.

To analyze correlation between the decay factor $\alpha$ and entropies, plots of the number of observations with a certain entropy are shown in Figure 3. In most games, higher decay factors increase the presence of low entropy states and decreases the presence of high entropy states. In the games with frequent reward occurences such as Enduro and MsPacman, there are more low-entropy observations than BeamRider and SpaceInvaders, where reward occurences are less frequent. Especially with regards to MsPacman, we believe that the TDW Average policy with larger $\alpha$ values results in worse performance because the agent frequently receives positive rewards at almost every timestep, which often induces prediction error $\delta^p$, and increases uncertainty in all Q-functions. Thus, globally weighted ensemble methods achieve better performance than TDW algorithms because it is difficult to consistently accumulate uncertainties on MsPacman.

## 6 CONCLUSION

In this paper, we have introduced the TDW algorithm: an ensemble method that accumulates temporal difference errors as an uncertainties in order to adjust weights of each Q-function, improving performance especially in high-dimensional state-space or situations where there are multiple optimal trajectories. We have shown performance evaluations in Gridworld tasks and Atari tasks, wherein the TDW algorithms have achieved significantly better performance than non-weighted algorithms and globally weighted algorithms. However, it is difficult to correctly measure uncertainties with frequent reward occurrences because the intrinsic prediction errors are also accumulated. Thus, these types of games did not realize the same performance improvements.

In future work, we intend to investigate an extension of this work into continuous action-space tasks because only the joint decision problem of Q-functions is considered in this paper. We believe a similar algorithm can extend a conventional ensemble method (Huang et al. (2017)) of Deep Deterministic Policy Gradients (Lillicrap et al. (2015)) by measuring uncertainties of pairs of a policy function and a Q-function. We will also consider a separate path, developing an algorithm that measures uncertainties without rewards because reward information is not always available especially in the case of real world application.

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

# A  Q-FUNCTION TABLES OBTAINED ON GRIDWORLDS

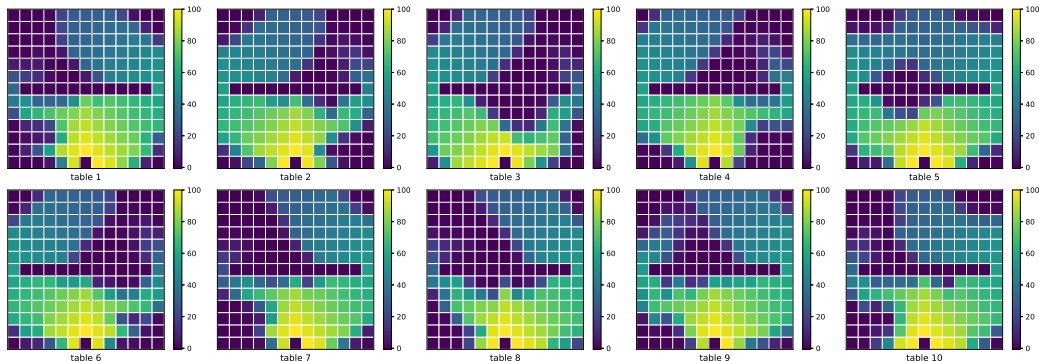

Figure 4: Learned table-based Q values at Two-slit Grid World. Each cell corresponds to state $s$, and its color represents a value of $\max_a Q(s, a)$.

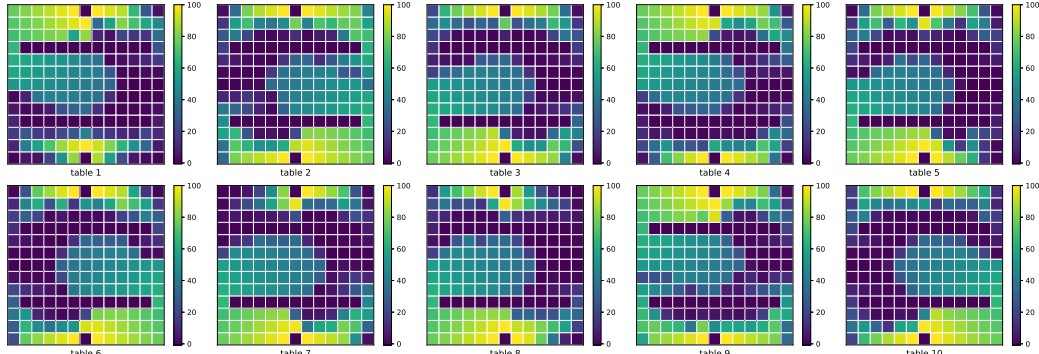

Figure 5: Learned table-based Q values at Four-slit Grid World. Each cell corresponds to state $s$, and its color represents a value of $\max_a Q(s, a)$.

