# OpenReview forum: "Temporal Difference Weighted Ensemble For Reinforcement Learning"
_ICLR.cc/2020/Conference — Reject_

### Official Review · AnonReviewer1 · 2019-10-23
**Official Blind Review #1**

**Rating:** 1

**Review:**

This paper addresses the problem of combining separately learned action-value functions into a single, no-longer-learning, action selection algorithm. The result is not necessarily a combined policy, strictly speaking, because it need not take the same action (or produce the same action selection probabilities) each time it is in the same state. Such “ensemble methods” have been a big topic in supervised learning, where ensemble methods are often effective, but less so in reinforcement learning, where the evidence in support of the idea is, perhaps, weak. This paper just assumes it is a good idea, while citing prior work that i don’t think really shows that it is. A new method is proposed, called TDW, based on weighting the learner’s action values according to their recent squared temporal-difference errors. This is compared with simple ensemble method that have no memory other than their action values, on two gridworld problems and six Atari games. Comparison is also made with single (non-ensemble) methods that are given as much training data as one of the members of the ensemble. The results are consistent with TDW being better than the simpler ensemble methods on both gridworlds and Atari games, and with both ensemble methods being better than the single methods on Atari games but worse than the single methods on the gridworld problems.

Does this work make a contribution to the field of reinforcement learning? A lot depends on whether you think the problem of combining separate learners is an important problem. I am skeptical. First, this problem requires one to be able to identify separate learners that are operating in independent copies of the identical environment, and whose states and actions can be mapped one-to-one onto each another. (This is also a requirement of asynchronous methods, so this criticism applies to them as well.) Of course it is easy for us to arrange these things with our simulated test environments, and there may even be _a few_ real problems like this, but most of the time this is unrealistic. If a method requires this, then it is not a general method. Second, and I suppose relatedly, this new problem bears little relation to the original problem: Does our field benefit from the introduction of new problems and solution methods that apply only in special cases? Or are these special cases primarily distractions from the general case and thus ways to avoid coming to grips with the real problem?

Different researchers might answer these rhetorical questions in different ways. It is a judgement call, but they reflect a real concern. A greater problem is that this work does not do a good job addressing the problem of ensembles. First, the introduction glosses over the question of whether ensembles are a good idea; it kinda suggests that their effectiveness has already been established, but does not commit itself to that statement; it does not make a clear claim that might be true or false about that prior work and that would validly support interest in this area. Second, this paper contains the wrong sort of comparisons of ensemble methods with single (non-ensemble) methods. It compares an ensemble of 10 learners with a single learner who only has as much data as one of the 10 learners in the ensemble. Thus, each single learner has only one-tenth as much data as has gone into the ensemble. Surely this is not the right comparison. It is a real problem that an early proponent of combining learners makes an unfair comparison with non-combined learners. It sets a poor standard and example for those who come later. The third way in which this work on ensemble RL agents is not very good it that its ensemble methods are not well suited to the task. That is, if one was going to combine learners, then there are many interesting natural ways to do this, and these are not done. It would be natural for each learner to keep track of how much experience or confidence it has in each part of the state space. The TDW algorithm can be seen as one way of doing this, but it is just one rather idiosyncratic way; there are many more natural ways. If we are going to explore this new problem, and new algorithms for solving it, then one really ought to do better than propose only methods that retain only their action-value function and nothing else that might facilitate the subsequent combination. In these three ways I feel this paper is not a good example of exploring the new problem of ensemble learning, even if you think that problem is a worthy one.


**Experience Assessment:**

I have published in this field for several years.

**Review Assessment: Checking Correctness Of Derivations And Theory:**

I assessed the sensibility of the derivations and theory.

**Review Assessment: Checking Correctness Of Experiments:**

I carefully checked the experiments.

**Review Assessment: Thoroughness In Paper Reading:**

I read the paper thoroughly.

---

> ### Author Response · Authors · 2019-11-08
> **Response to reviewer 1**
>
> We would like to sincerely thank you for the detailed comments. We addressed each question as follows. Please, let me organize the questions that appeared above.
>
> 1. Is arranging multiple members a realistic situation?
>
> We believe it is a very realistic and practical situation. In supervised learning, combining multiple classifiers to improve final performance is a frequently used technique. Even in reinforcement learning, in order to improve performance, it is natural to combine multiple members to create a single system. For example, when building autonomous driving cars with a deep reinforcement learning approach, improving final performance in the real environment can be done by combining multiple members individually trained in a simulator. We believe this is a very realistic situation and the place where our approach will be effectively applied.
>
> 2, Is this contribution to RL?
>
> We believe it is yes. Our approach leverages accumulated TD errors to adjust weights. The online error calculation a characteristic unique to the reinforcement learning setting. Thus, our approach will be very fundamental of ensemble methods of RL.
>
> 3. Is our setting a special case?
>
> We believe that our approach can be applied in general cases of RL. To show this, we conducted experiments on Atari games. As a result of the experiments, our approach generally improves performance to compare with simple ensemble techniques.
>
> 4. validity of ensemble methods in RL
>
> Our approach is inspired by the prior work [1] that drops unconfident members to improve ensemble performance. In their experiments, performance improvements are empirically shown and analyzed. We extend this idea to softly weighted ensemble method of RL. We also have empirically shown performance improvements on Atari tasks where each has different task settings, which suggests our approach is widely validated in the RL domain. However, we understand the importance of further theoretical analysis of our approach. We will investigate more of it in future work.
>
> 5. Is our comparison with the single learning the right way?
>
> In our experiments, we focus on improving performance from simple ensemble techniques. In this sense, we believe our comparison is done in the right way. Furthermore, in reinforcement learning, it is nearly impossible to give same amount of data to the single agent as the 10 learners have because each agent learns from data gathered by its self. In addition, to compare with prior works, our comparison is done in a natural way.
>
> 6. Is our approach a natural way?
>
> We believe our approach is a natural way of combining multiple members. Our approach does not require architecture changes nor additional components. Only online calculation of TD errors is enough to use our approach and it is simple enough to apply it to other deep reinforcement learning models.

---

> > ### Author Response · Authors · 2019-11-11
> > **Add reference**
> >
> > I'm sorry for undetailed reference above.
> >
> > Here is a reference.
> >
> > Reference
> > [1] Stefan Faußer and Friedhelm Schwenker. Selective neural network ensembles in reinforcement
> > learning: taking the advantage of many agents. Neurocomputing, 169:350–357, 2015b.

---

### Official Review · AnonReviewer4 · 2019-10-30
**Official Blind Review #4**

**Rating:** 3

**Review:**

1.	The paper suggests an ensemble method that leads to better performance over the execution of a single agent. The final (behavior) agent is a weighted average (or majority vote) of the members, where weights are determined by the accumulated TD error of the agent in the episode so far. The TD error serves as a confidence measure of the ensemble members in their predictions.
2.	Assume that one of the Q functions, Q_i is Q*. I.e., the optimal Q function. Let Q_ii = Q_i + B. I.e., a biased version of the optimal Q function. We know that both Q_i and Q_ii induce the same (optimal) policy. However, while delta_t^i is zero everywhere (the bellman error is zero for Q*), delta_t^ii is not zero (because of the discount factor). If B is large enough then delta_t^ii will be arbitrarily large to the point where the TDW algorithm will completely overlook Q_ii.
3.	Fear of uncertainty: in stochastic settings the TD error will almost surely won’t be zero. How does the method work in the presence of stochasticity?
4.	In my understanding, TDW favors policies that go to deterministic parts of the state space (where the TD error can be arbitrarily small), over policies that go to uncertain parts of the state space (where the TD error will never be zero). However, it is not difficult to construct an example where the optimal policy is to go to the uncertain parts of the state space. Therefore, TDW favors determinism over uncertainty. However, determinism is not necessarily linked to better performance.
5.	The paper links between certainty (as reflected in the TD error) and performance. If the performance criterion would be risk-sensitive, e.g. CVAR, then I could agree more with the claim (maybe then performance is linked with certainty). However, certainty and performance do not go together, at least not when talking about the expected return criteria.
6.	Out of curiosity: since the method requires calculating the max over Q. How would it work in a continuous action space?

**Experience Assessment:**

I have published one or two papers in this area.

**Review Assessment: Checking Correctness Of Derivations And Theory:**

I assessed the sensibility of the derivations and theory.

**Review Assessment: Checking Correctness Of Experiments:**

I did not assess the experiments.

**Review Assessment: Thoroughness In Paper Reading:**

I read the paper at least twice and used my best judgement in assessing the paper.

---

> ### Author Response · Authors · 2019-11-08
> **Response to reviewer 4**
>
> We would like to sincerely thank you for the detailed comments. We addressed each question as follows.
>
> 3. "Fear of uncertainty: in stochastic settings the TD error will almost surely won’t be zero. How does the method work in the presence of stochasticity?"
>
> We believe that stochasticity is a very important topic in RL. As all other researches of deep reinforcement learning, we first try the deterministic environments to formalize our approach. In future work, we definitely will consider extensions to stochastic environments.
>
> 4, 5.  "However, determinism is not necessarily linked to better performance." & "However, certainty and performance do not go together, at least not when talking about the expected return criteria."
>
> In our paper, we call accumulated TD errors "uncertainty". In the context of function approximation, the member with large uncertainty cannot approximate values correctly under current state transitions. Our approach is giving small weights to such members to improve performance. A prior work [1] cited in our paper shows that dropping members with large TD errors improves performance because the members that cannot predict values correctly would have a bad influence on joint action decisions. We extend this idea to a softly weighted ensemble method. Therefore, in order to improve performance, we believe that giving small weights to unconfident members makes sense.
>
> 6. "How would it work in a continuous action space?"
>
> I appreciate this question. In deep deterministic policy gradients (DDPG) [2], using the policy function $a=\pi(s)$, the max $Q$ is calculated as $\max Q = Q(s, \pi(s))$. We will use this formulation in our future work.
>
> Reference
> [1] Stefan Faußer and Friedhelm Schwenker. Selective neural network ensembles in reinforcement
> learning: taking the advantage of many agents. Neurocomputing, 169:350–357, 2015b.
> [2] Timothy P Lillicrap, Jonathan J Hunt, Alexander Pritzel, Nicolas Heess, Tom Erez, Yuval Tassa,
> David Silver, and Daan Wierstra. Continuous control with deep reinforcement learning. arXiv
> preprint arXiv:1509.02971, 2015.

---

### Official Review · AnonReviewer5 · 2019-11-01
**Official Blind Review #5**

**Rating:** 8

**Review:**

Overall, this is a very interesting paper and, I think, would make a great addition to ICLR.  I find the ideas discussed in the paper to be stimulating and important to the community.

The authors are addressing a frequently overlooked and under-discussed aspect of RL: namely the instability of training that occurs when a Q-function becomes decreasingly familiar with areas of a state space while it focuses on a particular "trajectory" or solution to a task.

The authors add a measurement of uncertainty which is a simple heuristic.  It does require a couple additional hyperparameters.  One of which is not investigated in the experiments.

The experiments are ample and the approach is compared to a baseline and an alternative approach.  The experiments are informative and show the superiority of the proposed approach.

Comments:

The paper initially left me with the impression that the members were trained separately but Algorithm 1 looks like they are trained together.  This would have a significant impact on the number of training time steps required for training.  Please clarify.

The literature review is good but must add two relevant works.  One is MMRL by Doya et al.  I think their notion of a forward model is similar in spirit so it should be discussed.  Also, see the dissertation by DL Elliott entitled: THE WISDOM OF THE CROWD: RELIABLE DEEP REINFORCEMENT LEARNING THROUGH ENSEMBLES OF Q-FUNCTIONS.

Section 3.3 didn't elucidate the topic for me.  It could be removed.  I think the description of the algorithm is sufficient.  Could be replaced by additional analysis of the uncertainty parameter from the experiments.  Also, what is the importance of the variable L in (5)?

What does i denote in (6)?  I assume it's the index of the ensemble member.  Also, mention how the uncertainty values are initialized here.

I feel that the paper is never really solidified in the mind until seeing (7).

Your experiments indicate that the uncertainty value causes the "preferred" ensemble member to switch during the game of breakout.  That makes a lot of sense given the cyclical, for lack of a better term, nature of the game.  How about for tasks like the bipedal walker or the cart-pole task?  Would you expect, when training is completed, for a single member to dominant from the start position to the end of the trial?  If so, that's a limitation of the approach that doesn't diminish the contribution in my opinion.

Would be interesting to see how the number of ensemble members used during a trial changes during training.  Does it increase/decrease?  Would love to see more analysis of this.  AGAIN, I don't think it's a negative thing if all are used but, eventually, one dominates.

How would using simpler Q-function approximators change the results?  Would it force them to decompose the task?

**Experience Assessment:**

I have published in this field for several years.

**Review Assessment: Checking Correctness Of Derivations And Theory:**

N/A

**Review Assessment: Checking Correctness Of Experiments:**

I carefully checked the experiments.

**Review Assessment: Thoroughness In Paper Reading:**

I read the paper thoroughly.

---

> ### Author Response · Authors · 2019-11-08
> **Response to reviewer 5**
>
> We would like to sincerely thank you for the detailed comments. We addressed each question as follows.
>
> 1.  "The paper initially left me with the impression that the members were trained separately but Algorithm 1 looks like they are trained together."
>
> We believe we tried to clarify this in the first paragraph of the Introduction. Each member is trained individually. At evaluation time, all the members join action decisions without further training. I'll add more descriptions to the revised version.
>
> 2. "The literature review is good but must add two relevant works"
>
> Thank you for your suggestions. I'll take them into the section 2.
>
> 3. "Section 3.3 didn't elucidate the topic for me"
>
> I'll appreciate your pointing it out. I agree with the idea of its replacement with additional analysis. I'll revise the paper based on this.
>
> 4. "What does i denote in (6)?  I assume it's the index of the ensemble member.  Also, mention how the uncertainty values are initialized here."
>
> $i$ denotes its index of the member, which is already described in section 3.2. The uncertainty is initialized by 0.0 at each episode. I'll add a line of this into Algorithm 1.
>
> 5. "I feel that the paper is never really solidified in the mind until seeing (7)."
>
> I appreciate this indication. I'll add more descriptions to the introduction.
>
> 6. "How about for tasks like the bipedal walker or the cart-pole task? "
>
> As each member is individually trained, which captures slightly different policies with different seed values, the large weights would be always given to different members with random start. However, the weights would be smaller than the ones in Atari tasks because cart-pole task has significantly small state-space to compare with Atari.
>
> 7. "Would be interesting to see how the number of ensemble members used during a trial changes during training"
>
> We believe this is analyzed at Figure 4. As each contribution is softly combined with the weights, we cannot say the exact number of members that dominate action decisions. However, we measure this as entropy of the weights.
>
> 8. "How would using simpler Q-function approximators change the results?  Would it force them to decompose the task?"
>
> We believe this is one of our future directions. We expect it is possible to decompose large tasks into small ones by applying our method during training.

---

### Comment · Area_Chair1 · 2019-11-13
**Thanks for your reviews. Please take a look at the rebuttal.**

Dear reviewers,

Thank you very much for your efforts in reviewing this paper.

The authors have provided their rebuttal. It would be great if you take a look at them, and see whether it changes your opinion in anyway. If there is still any unclear point or a serious disagreement, please bring it up. Also if you are hoping to see a specific change or clarification in the paper before you update your score, please mention it.

The authors have only until November 15th to reply back.

I also encourage you to take a look at each others’ reviews. There might be a remark in other reviews that changes your opinion.

Thank you,
Area Chair

---

### Author Response · Authors · 2019-11-15
**Paper revision submitted**

I'd like to announce the paper revision uploaded. I appreciate all the reviewers' works. Let me list the changes below.

1. add minor changes to introduction and atari experiment
2. remove section 3.3
3. add a new citation of Doya et al. 2002 to section 2


Best regards.

---

### Decision · Program_Chairs · 2019-12-19

**Decision:**

Reject

**Comment:**

The paper proposes a method to combine the decision of an ensemble of RL agents. It uses an uncertainty measure based on the TD error, and suggests a weighted average or weighted voting mechanism to combine their policy or value functions to come up with a joint decision.
The reviewers raised several concerns, including whether the method works in the stochastic setting, whether it favours deterministic parts of the state space, its sensitivity to bias, and unfair comparison to a single agent setting.
There is also a relevant PhD dissertation (Elliot, 2017), which the authors surprisingly refused to discuss and cite because apparently it was not published at any conference. A PhD dissertation is a citable reference, if it is relevant. If it is, a good scholarship requires proper citation.

Overall, even though the proposed method might potentially be useful, it requires further investigations. Two out of three reviewers are not positive about the paper in its current form. Therefore, I cannot recommend acceptance at this stage.

Elliott, Daniel L., The Wisdom of the crowd : reliable deep reinforcement learning through ensembles of Q-functions, PhD Dissertation, Colorado State University, 2017